# TRIM21 Expression as a Prognostic Biomarker for Progression-Free Survival in HNSCC

**DOI:** 10.3390/ijms24065140

**Published:** 2023-03-07

**Authors:** Amelie von Bernuth, Julika Ribbat-Idel, Luise Klapper, Tobias Jagomast, Dirk Rades, Anke Leichtle, Ralph Pries, Karl-Ludwig Bruchhage, Sven Perner, Anne Offermann, Verena Sailer, Christian Idel

**Affiliations:** 1Institute of Pathology, University of Luebeck, University Hospital Schleswig-Holstein, Campus Luebeck, Ratzeburger Allee 160, 23538 Luebeck, Germany; 2Department of Radiation Oncology, University of Luebeck, University Hospital Schleswig-Holstein, Campus Luebeck, Ratzeburger Allee 160, 23538 Luebeck, Germany; 3Department of Otorhinolaryngology, University of Luebeck, University Hospital Schleswig-Holstein, Campus Luebeck, Ratzeburger Allee 160, 23538 Luebeck, Germany; 4Pathology, Research Center Borstel, Leibniz Lung Center, Parkallee 1-40, 23845 Borstel, Germany; 5Institute of Hematopathology & Medical Care Center Hanse Histologikum, Fangdieckstr. 75a, 22547 Hamburg, Germany

**Keywords:** TRIM21, head and neck squamous cell carcinoma (HNSCC), prognostic biomarker, immune cell infiltration, IHC

## Abstract

Patients with head and neck squamous cell carcinoma (HNSCC) continue to have a rather poor prognosis. Treatment-related comorbidities have negative impacts on their quality of life. TRIM21 is a cytosolic E3 ubiquitin ligase that was initially described as an autoantigen in autoimmune diseases and later associated with the intracellular antiviral response. Here, we investigated the role of TRIM21 as a biomarker candidate for HNSCC in predicting tumor progression and patient survival. We analyzed TRIM21 expression and its association with clinical-pathological parameters in our HNSCC cohort using immunohistochemistry. Our HNSCC cohort included samples from 419 patients consisting of primary tumors (*n* = 337), lymph node metastases (*n* = 156), recurrent tumors (*n* = 54) and distant metastases (*n* = 16). We found that cytoplasmic TRIM21 expression was associated with the infiltration of immune cells into primary tumors. In addition, TRIM21 expression was significantly higher in primary tumors than in lymph node metastases, and increased TRIM21 expression was correlated with shorter progression-free survival in HNSCC patients. These results suggest that TRIM21 could be a new biomarker for progression-free survival.

## 1. Introduction

Head and neck squamous cell carcinomas (HNSCCs) are epithelial malignancies that arise from mucosal surfaces of the oral cavity, pharynx, and larynx. HNSCCs are generally associated with a poor prognosis, as shown by the development of recurrences in over 50% of patients [1,2,3]. Risk factors for the development of HNSCC include behavioral risk factors, such as alcohol consumption and smoking, and human papillomavirus (HPV) infection [4]. Men have a higher incidence of HNSCC than women, which is thought to be due to higher exposure to behavioral risk factors [5,6]. The current therapies for HNSCC include chemotherapy, radiotherapy, surgery, and (only recently) immune checkpoint inhibitors for recurrent tumors [7]. These treatments can have severe side effects that impair patients’ quality of life. For instance, surgery may result in functional and visible impairments and radiotherapy may result in dysphagia, dental decay, and altered taste [8].

Identifying biomarkers for lower versus higher risk subgroups is important, as this could reduce the aggressiveness of the chosen treatment approach, i.e., help reduce the intensity of systemic treatment in low-risk groups and ultimately improve survivors’ quality of life. For oropharyngeal carcinoma, p16 is associated with a better prognosis and is currently used as a surrogate marker for HPV infection in the Fourth WHO Classification of Head and Neck Tumors [9,10]. However, the reliability of the p16-IHC test as a surrogate marker for HPV status is controversial [11]. Furthermore, immune evasion is a pathological finding in tumor specimens associated with a poor prognosis [12,13]. Nevertheless, additional biomarkers for HNSCC are needed to predict its prognosis, i.e., tumor recurrence, progression, and patient survival.

Recently, tripartite motif containing-21 or TRIM21 has been identified to be a possible HNSCC prognostic marker [14]. TRIM21 is a cytosolic E3 ubiquitin ligase that was initially described as an autoantigen in autoimmune diseases and later associated with the intracellular antiviral response [15,16]. TRIM21 induces the ubiquitin-mediated proteasomal degradation of proteins. Furthermore, TRIM21 binds with cytosolic Fc-receptor antibody–virus-complexes, leading to the degradation of the former and the upregulation of inflammatory signaling pathways [17].

Here, we analyzed the implementation of TRIM21 as a prognostic marker in a large human cohort of HNSCC patients.

## 2. Results

### 2.1. Patient Characteristics

The HNSCC cohort included samples of primary tumors (PTs; *n* = 337), lymph node metastases (LMs; *n* = 156), recurrent tumors (RTs; *n* = 54), and distant metastases (DMs; *n* = 16). The anatomical sites of PTs were the oral cavity (*n* = 78), larynx (*n* = 95), hypopharynx (*n* = 46), and oropharynx (*n* = 107). The cohort consisted of 89 female (23%) and 306 male (77%) patients. The age of the patients ranged from 29 to 90 years (mean 62.5 years). Using p16 as a surrogate marker for HPV infection, 116 (29%) patients were classified as HPV-positive. Most patients received upfront surgery (98%), and 2% of our patients received salvage surgery. In addition, the patients received either radiotherapy (40%) and/or chemotherapy (30%). The results of our analysis are summarized in Table A1.

### 2.2. TRIM21 Staining Patterns in Our HNSCC Cohort

We observed the cytoplasmic staining of TRIM21 in tumor cells (Figure 1). Our cohort’s intratumoral staining patterns were heterogeneous, meaning we observed both negative and positive cells in the same tumor sample. We used the mean positive index for each sample to adequately address these patterns. The mean positive index was considered a measure of protein expression.

### 2.3. TRIM21 Expression Is Higher in Primary Tumors Compared with Lymph Node Metastases

We compared the positive index of the cytoplasmatic TRIM21 expression in samples from PTs with that of RTs, LMs, and DMs (Figure 2). The positive index of PTs vs. RTs (Mann–Whitney test, *p* = 0.551) and PTs vs. DMs (Mann–Whitney test, *p* = 0.801) did not significantly differ. The positive index was significantly higher in PTs than in LMs (Mann–Whitney test, *p* < 0.001).

Furthermore, we matched the PT samples (*n* = 123) with the corresponding LM samples from the same patients (Figure 3). In the matched-pair analysis of PTs and the corresponding LMs, the difference in the TRIM21 remained significant (paired sample *t*-test, *p* = 0.007).

### 2.4. TRIM21 Expression Regarding the Anatomical Site of the Primary Tumor

We compared the TRIM21 expression in PTs regarding their anatomical site of origin, i.e., oral cavity, hypopharynx, oropharynx, or larynx. No significant differences were observed in the positive index regarding PT location (Kruskal–Wallis test, *p* = 0.358) (Figure 4).

### 2.5. TRIM21 Expression Is Higher in Primary Tumors with Immune Infiltration

We assessed the positive index of PTs with immune cell infiltration and PTs without immune cell infiltration. Hot and excluded PTs (*n* = 213) had a significantly higher positive index (Mann–Whitney test, *p* = 0.008) than cold PTs (*n* = 65). When comparing hot (*n* = 66) vs. excluded (*n* = 147) PTs, we did not observe a significant difference (Mann–Whitney test, *p* = 0.349). In hot vs. cold (Mann–Whitney test, *p* = 0.011) and excluded vs. cold (Mann–Whitney test, *p* = 0.020) tumors, the TRIM21 expression was significantly higher in hot and excluded tumors (Figure 5).

We also compared the positive index of RTs with immune cell infiltration vs. RTs without immune cell infiltration. We did not observe a significant difference (Mann– Whitney test, *p* = 0.239) between hot/excluded (*n* = 16) and cold (*n* = 16) RTs. Nevertheless, the trend observed for PTs was also observed for RTs.

### 2.6. Higher TRIM21 Expression Is Associated with a Shorter Progression-Free, but Not with a Shorter Overall Survival

We evaluated the prognostic value of TRIM21 expression for overall survival (OS) and progression-free survival (PFS) in HNSCC patients. We determined the optimal cut-off for TRIM21 expression to predict PFS and OS at 60 months through receiver operating characteristic (ROC) curve analysis. Using the cut-off of 0.37, we created two groups (high: TRIM21 expression above cut-off, *n* = 246; low: TRIM21 expression below cut-off, *n* = 80). In the Kaplan–Meier survival curve and log-rank test, we recorded statistically significant shorter PFS rates in HNSCC patients with high TRIM21 expression compared with HNSCC patients with low TRIM21 expression (log-rank test, *p* = 0.024) (Figure 6). The 5-year PFS rates were estimated at 45.9% for high TRIM21 expression and 59.4% for low TRIM21 expression. Subsequently, a univariable and multivariable Cox regression analysis was performed to confirm whether the prognostic value of the TRIM21 expression was independent of p16 status, UICC stage, T stage, and N stage. Univariate Cox regression analysis revealed a significant correlation between patients’ PFS and TRIM21 expression (hazard ratio (HR) = 0.63 (95% CI 0.41–0.97), *p* = 0.035). High TRIM21 expression remained associated with a significantly shorter PFS in multivariate Cox regression and therefore predicted PFS independently of the factors listed above (HR = 0.63 (95% CI 0.41–0.98), *p* = 0.040). In addition, we identified p16 status, UICC stage, T stage, and N stage as independent prognostic factors for 5-year PFS in our cohort. The results of our analysis are summarized in Table 1. However, we did not observe a significant correlation between OS and TRIM21 expression in our cohort (log-rank test, *p* = 0.81) (Figure 6). These results indicate that TRIM21 may be an independent predictor for PFS in HNSCC patients.

### 2.7. TRIM21 Expression Regarding Clinicopathological Features of HNSCC Patients

The correlation of TRIM21 expression with the clinicopathological characteristics of patients was analyzed using the following parameters: age, sex, grading, occurrence of DMs, occurrence of RT, alcohol abuse, nicotine consumption, p16 status, UICC stage, T stage, and N stage at initial diagnosis. However, no significant results were observed. Table 2 summarizes our findings.

## 3. Discussion

A recent publication identified TRIM21 as a possible prognostic biomarker for the OS of HNSCC patients [14]. Higher TRIM21 mRNA expression is correlated with a shorter OS according to Gene Expression Profiling Interactive Analysis (GEPIA) data. Here, we reported on TRIM21 protein expression and its relationship to clinicopathological parameters in a large HNSCC cohort to evaluate the role of TRIM21 as a candidate biomarker for HNSCC.

Our comparison of TRIM21 expression in different tissue types revealed a significantly higher level of TRIM21 expression in PTs compared with LMs. In ovarian, renal, and hepatocellular carcinoma, TRIM21 overexpression inhibits cell migration in vitro and in vivo [18,19,20]. Nevertheless, we did not observe significant differences in the TRIM21 expression of PTs in terms of the occurrence of LMs. This may be an indication that the tumor biology of LMs differs from that of accompanying PTs.

Furthermore, we observed significantly higher TRIM21 expression in PTs with immune cell infiltration compared with PTs without immune cell infiltration. Sjöstrand et al. (2013) reported that TRIM21 expression is upregulated by the IFN/JAK/STAT signaling pathway during viral infections [21]. In addition, the JAK/STAT signaling pathway has been found to be constitutively activated in HNSCC [22]. Moreover, TRIM21 is able to induce signaling cascades that lead, for instance, to the activation of inflammatory cytokine production [23]. Nevertheless, we cannot make assumptions about the development of TRIM21 expression during tumor progression, i.e., whether carcinogenesis leads to increased TRIM21 expression and thus to increased immune cell infiltration or whether immune cell infiltration leads to increased TRIM21 expression. Therefore, functional analyses are needed to better understand the relationship between immune cell infiltration and TRIM21 expression in HNSCC.

We also observed increased TRIM21 expression in RTs with immune cell infiltration relative to those without, but the difference was not statistically significant. This could have been due to the small sample size in our cohort of RTs. In future studies, whether TRIM21 expression differs in RTs regarding immune cell infiltration status needs to be investigated.

The role of TRIM21 as a prognostic biomarker has previously been analyzed in other cancer entities. For example, decreased TRIM21 expression in ovarian cancer, diffuse large cell lymphoma, and breast cancer is associated with shorter OS rates [18,24,25]. In thyroid cancer, higher TRIM21 expression is correlated with an increased risk of recurrences and lymph node metastases [26]. In contrast, increased TRIM21 expression in glioma and hepatocellular carcinomas is associated with a poorer prognosis [27,28]. We found that a higher level of TRIM21 expression was associated with shorter PFS rates in our cohort. However, we did not observe an association between increased TRIM21 expression and a shorter OS, though previous studies have reported on increased mortality rates of HNSCC patients. In addition to PTs, causes of death include new malignant tumors and non-cancer causes such as treatment-related and alcohol- and tobacco-associated comorbidities [29,30]. From the available data in our cohort, we were only able to determine OS and not disease-specific mortality. In order to evaluate the value of TRIM21 as a prognostic marker, disease-specific mortality, in particular, should be further investigated.

In nasopharyngeal cancer (NPC), TRIM21 has been reported to promote the radiation resistance of NPC cells [31]. Through the ubiquitination and degradation of guanine monophosphate synthase (GMPS), TRIM21 suppresses TP53 expression. Nevertheless, NPC differs from other HNSCCs in, for example, its carcinogenesis [32]. Future research could investigate whether TRIM21 also plays a role in the development of resistance after radiotherapy in other HNSCC subsites.

The HPV E7 oncoprotein has been described to activate the TRIM21-mediated proteasomal degradation of gamma-interferon-inducible protein-16 (IFI16) in cervical cancer cell lines. The decrease in IFI16 leads to the disruption of pyroptosis and thus enables persistent HPV infections. We hypothesized that the TRIM21 expression might be increased in HPV-positive tumors, but we did not observe a significantly higher TRIM21 expression in HPV-positive tumors. One possible explanation for this could be that the average latency between HPV infection and cancer is at least 10 years [33]. Due to the lack of molecular biomarkers to identify premalignant HNSCC lesions and predict their progression, it may be interesting to investigate the role of TRIM21 in the carcinogenesis of HPV-positive tumors.

In conclusion, cytoplasmatic TRIM21 expression was found to be associated with immune cell infiltration in PTs, as well as a worse PFS rates in HNSCC patients. Furthermore, LMs showed lower TRIM21 expression levels than PTs.

TRIM21 could serve as a new prognostic biomarker for disease progression in HNSCC patients. However, the prognostic value of TRIM21 needs to be validated in other cohorts.

## 4. Materials and Methods

### 4.1. Patients and HNSCC Cancer Specimens

We analyzed TRIM21 expression in HNSCC specimens derived from a cohort of 419 patients diagnosed between 2012 and 2015 at the Institute of Pathology of the University Medical Center Schleswig-Holstein, Germany, as reported previously [13,34,35,36,37,38,39]. Tumor samples were derived from PTs, LMs, RTs and DMs. We excluded patients with nasopharyngeal cancer from our cohort since the epidemiology, carcinogenesis, and treatment of nasopharyngeal tumors differ from those of other HNSCCs. Characteristics and clinical data were obtained through a review of clinical records and pathology reports. We defined OS as the duration from the date of initial diagnosis until death, regardless of the cause of death. For patients that survived longer than the cut-off of 60 months after diagnosis, the OS time was set to 60 months. PFS was defined as the length of time between the initial diagnosis and the diagnosis of RT or death, regardless of the cause of death. For cases that survived longer than or had no RT prior to the cut-off of 60 months after diagnosis, the PFS time was set to 60 months. We classified tumors according to the eighth edition of UICC TNM classification [40].

Ethical approval was obtained from the Ethics Committee of the University of Luebeck (AZ16-277). The samples were collected in accordance with the Declaration of Helsinki. All patients consented to the use of their tissue and data for research.

### 4.2. Tissue Microarray Construction

Formalin-fixed paraffin-embedded (FFPE) HNSCC tissue samples (4 μm thick) were mounted on slides and stained with hematoxylin and eosin (H&E) using standard protocols, as reported previously [41]. The tumor region was subsequently identified on the slides and marked on the FFPE blocks. Three 1 mm^2^ cores were obtained for every cancer sample and arranged on a tissue microarray (TMA) using a semiautomatic tissue arrayer (Beecher Instruments, Sun Prairie, WI, USA), as previously described [42]. Each TMA consisted of up to 54 tumor samples, up to 6 samples of benign tissue from the head and neck region, and 3 liver samples for orientation.

### 4.3. Immunohistochemistry

TRIM21 expression in HNSCC cells was assessed using immunohistochemistry. After the deparaffinization of the FFPE tumor tissues and heat-mediated antigen retrieval, immunohistochemical staining was performed as previously described [41]. We applied the polyclonal rabbit anti-human TRIM21/RO52 antibody (1:100, IHC, Cat# LS B15291 LifeSpan BioSciences, Seattle, WA, USA) and revealed its binding through automated IView DAB Detection using a Ventana BenchMark automated staining system (Roche, Basel, Switzerland), as described previously [42]. Human liver tissue served as a positive control, and lymphocytes served as a negative control.

### 4.4. Digitalization and Evaluation

The scanning and digitalization of the stained slides were performed using a Ventana iScan HT scanner (VentanaTuscon, AZ, USA) with a 40-fold objective. The software QuPath (version 0.2.3) was used to assess the digitalized files [43]. TMA cores were identified using QuPath’s TMA dearranger function, and tumor areas were manually annotated. Cytoplasmic immunoreactivity was expressed as the positive index, meaning the ratio number of positively stained cells divided by the total number of cells. Parameters for cell detection and thresholds for classification in negative and positive cells were verified by a board-certified pathologist. Positive cells within annotated areas were counted using the Positive Cell Detection command. A script was generated and run on all individual TMA slides to automate the detection process. The mean value of the available cores was calculated for each tissue sample for statistical analyses.

### 4.5. Immune Cell Infiltration

To assess the immune cell infiltration status of the PTs and RTs, H&E-stained tumor sections were classified by a board-certified pathologist into 3 categories according to the presence or absence of tumor-infiltrating immune cells. Diffusely immune-infiltrated tumors were described as “hot”, tumors with immune cell infiltration restricted to stromal areas were described as “excluded”, and tumors without immune cell infiltration were described as “cold” [13].

### 4.6. P16 Status

To assess the HPV status in the tumor samples, we used the p16 status as a surrogate marker [9]. Protein expression was detected with immunohistochemical staining using the mouse monoclonal antibody p16 (p16 CINtec ready to use kit, clone E6H4™, Roche Ventana Medical Systems, Tucson, AZ, USA), as described previously [18,38]. Binding was revealed using the Dab system delineated above.

### 4.7. Statistical Analysis and Visualization

Statistical evaluation was carried out using the software jamovi (Version 2.3. accessed on 11 April 2022) [44], which is built on top of the R statistical language, and the R packages finalfit and survival [45,46,47]. First, the conformation of the data to a normal distribution was determined using the Shapiro–Wilk test; consequently, a suitable statistical method was selected. The correlation of TRIM21 expression with the clinicopathological features of patients and different tissue types was analyzed using the Mann–Whitney test. TRIM21 expression levels at different anatomical sites of PTs were compared with the Wilcoxon rank test. The comparison of TRIM21 expression in matched PTs and LMs was performed using the paired *t*-test. The optimal cut-off for the TRIM21 expression to predict PFS and OS at 60 months was determined using ROC curve analysis and maximizing the sum of sensitivity and specificity. The Kaplan–Meier method and log-rank tests were used to calculate 5-year OS and PFS probability and to test for statistical significance. Univariate and multivariate survival analyses were performed for the PFS using the Cox proportional hazards regression model. The significance level was set at *p* < 0.05. All tests were two-sided.

Box plots were plotted with the R tidyverse and ggsignif packages, and survival curves were fitted with the R survminer and ggfortify packages [48,49,50,51]. We used Microsoft PowerPoint (Microsoft Office 365 ProPlus, Microsoft, Redmont, Washington, DC, USA) and ImageJ to edit pictures [52].

## Figures and Tables

**Figure 1 ijms-24-05140-f001:**
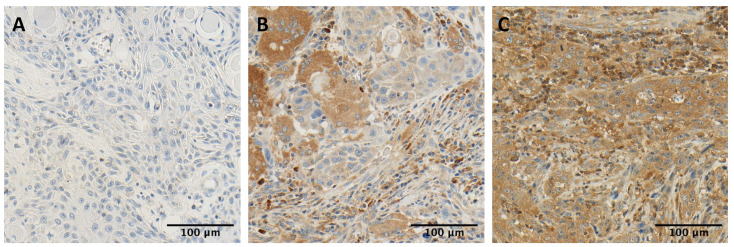
Staining patterns for TRIM21 in HNSCC. We evaluated TRIM21 expression in HNSCC samples from our cohort using immunohistochemistry. We observed heterogenous intratumoral staining patterns, meaning both positive and negative cells were observed within the patients’ samples. In (**A**), tumor cells show no positive staining with TRIM21. In (**B**,**C**), the tumor cells show a positive cytoplasmatic staining with TRIM21.

**Figure 2 ijms-24-05140-f002:**
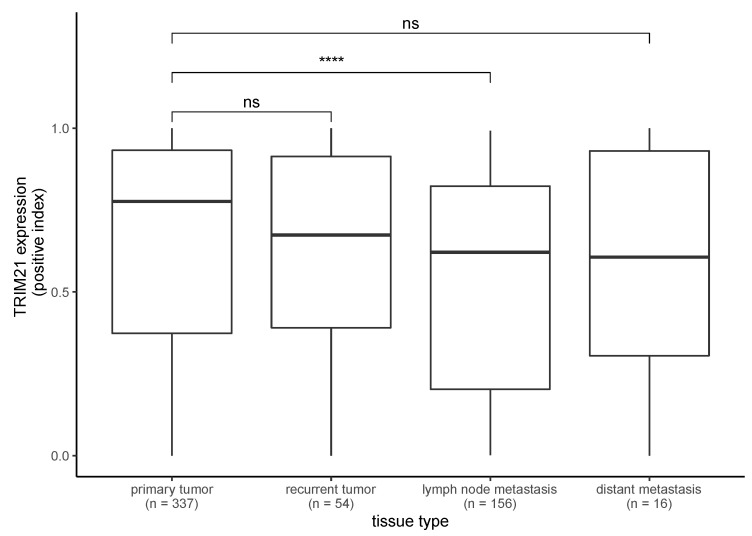
Expression of TRIM21 in PTs compared with LMs, RTs, and DMs. TRIM21 expression in PTs was significantly higher compared with LMs (Mann–Whitney test, *p* < 0.001). Other comparisons did not reach significance (Mann–Whitney test, *p* > 0.05). (**** *p* < 0.001, ns = not significant).

**Figure 3 ijms-24-05140-f003:**
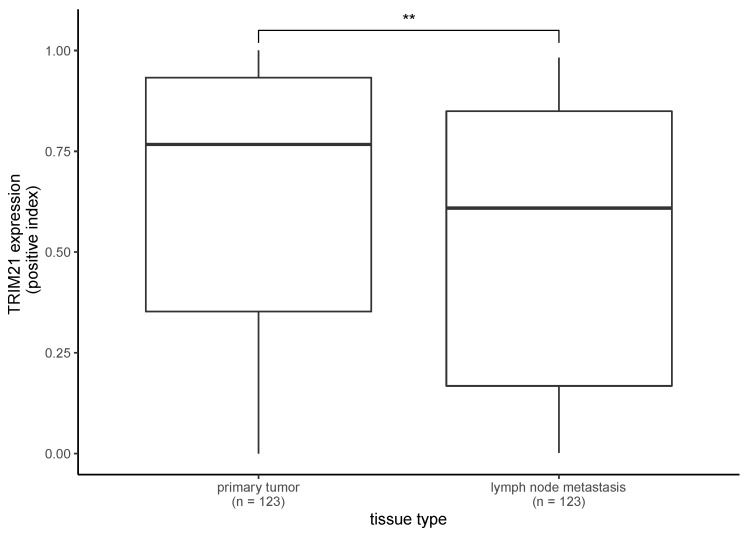
TRIM21 expression in primary tumors matched with the corresponding lymph node metastases from the same patient. TRIM21 expression in primary tumors and lymph node metastases also significantly differed in matched samples (paired sample *t*-test, *p* = 0.007). (** *p* < 0.01).

**Figure 4 ijms-24-05140-f004:**
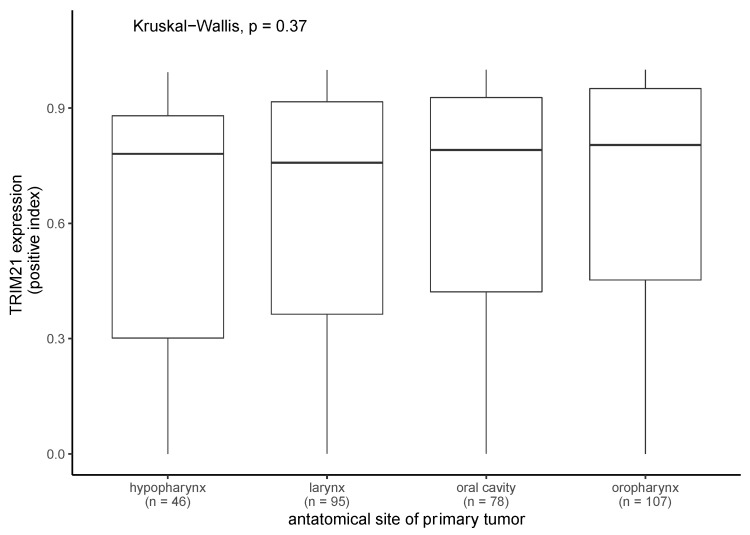
TRIM21 expression in primary tumors regarding their anatomical site of origin. TRIM21 expression in primary tumors did not significantly differ in regard to their anatomical site of origin (Kruskal–Wallis test, *p* = 0.37).

**Figure 5 ijms-24-05140-f005:**
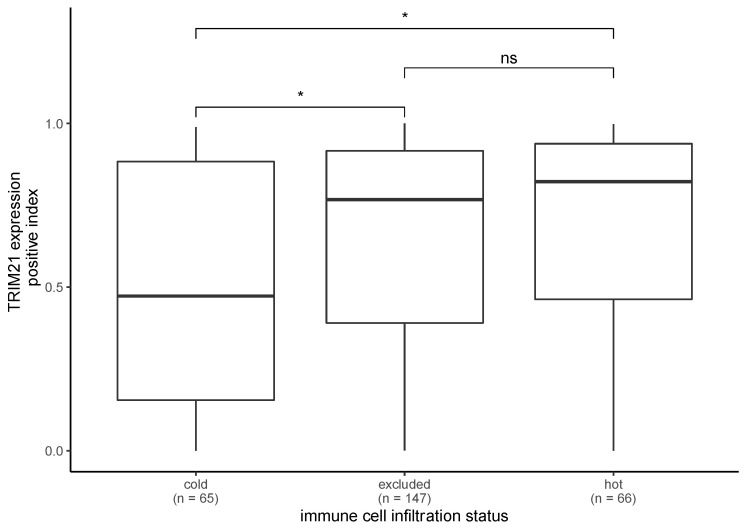
TRIM21 expression in primary tumors regarding immune cell infiltration. There were significant differences in the TRIM21 expression between cold and excluded primary tumors (Mann–Whitney test, *p* = 0.02) and between hot and cold primary tumors (Mann–Whitney test, *p* = 0.011). (* *p* < 0.05, ns = not significant).

**Figure 6 ijms-24-05140-f006:**
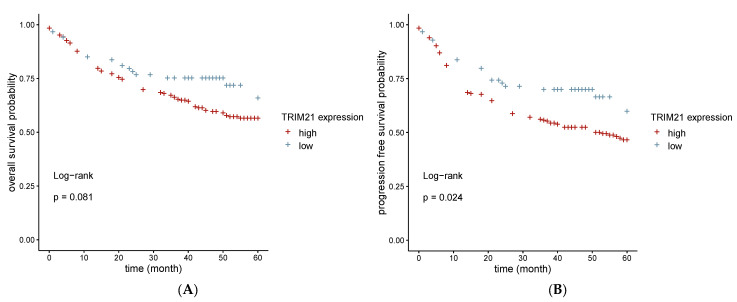
Kaplan–Meier analyses of TRIM21 expression in PTs. (**A**) We did not observe a significant difference in OS over 60 months in patients with high TRIM21 expression compared with patients with low TRIM21 expression (log-rank test, *p* = 0.081). (**B**) We observed a significantly shorter 5-year PFS rate in patients with high TRIM21 expression compared with patients with low TRIM21 expression (log-rank test, *p* = 0.024).

**Table 1 ijms-24-05140-t001:** Cox regression analysis of TRIM21 expression (* *p* < 0.05).

Clinicopathological Variable	Univariate Survival Analysis		Multivariate Survival Analysis	
	HR (95% CI)	*p*-Value	HR (95% CI)	*p*-Value
TRIM21 expression High (*n* = 242) Low (*n* = 79)	0.63 (0.41–0.97)	0.035 *	0.63 (0.41–0.98)	0.040 *
p16 Negative (*n* = 237) Positive (*n* = 84)	0.42 (0.27–0.67)	<0.001 *	0.44 (0.26–0.73)	0.002 *
UICC UICC I and II (*n* = 124) UICC III and IV (*n* = 197)	2.31 (1.59–3.34)	<0.001 *	1.07 (0.59–1.93)	0.829
T stage T1 and T2 (*n* = 166) T3 and T4 (*n* = 155)	2.17 (1.55–3.03)	<0.001 *	1.80 (1.14–2.83)	0.011 *
N stage N0 (*n* = 142) Nx (*n* = 179)	1.42 (1.01–1.98)	0.042 *	1.44 (0.97–2.14)	0.069

**Table 2 ijms-24-05140-t002:** TRIM21 expression in primary tumors regarding different clinicopathological features. *p*-values were calculated using the Mann–Whitney test.

Clinicopathological Features	*n*	*p*-Value
Sex	female (*n* = 74) vs. male (*n* = 262)	0.142
Age	≤62 years old (*n* = 168) vs. >62 years old (*n* = 167) *	0.295
Alcohol abuse	no (*n* = 183) vs. yes (*n* = 141)	0.174
Nicotine consumption	no (*n* = 36) vs. yes (*n* = 283)	0.143
p16 status	negative (*n* = 250) vs. positive (*n* = 87)	0.264
UICC stage at initial diagnosis	UICC I and II (*n* = 128) vs. UICC III and IV (*n* = 208)	0.582
T stage at initial diagnosis	T1 and T2 (*n* = 171) vs. T3 and T4 (*n* = 164)	0.242
N stage at initial diagnosis	N0 (*n* = 147) vs. Nx (*n* = 187)	0.147
Distant metastasis	no (*n* = 291) vs. yes (*n* = 45)	0.490
Grading	G 1 and G 2 (*n* = 261) vs. G 3 (*n* = 73)	0.252
Recurrence	no (*n* = 263) vs. yes (*n* = 74)	0.061

* Age grouping was performed according to the median.

## Data Availability

The data presented in this study are available on request from the corresponding author.

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
