# Peer review of "TRIM21 Expression as a Prognostic Biomarker for Progression-Free Survival in HNSCC"

_ijms, 2023, doi:10.3390/ijms24065140_

Round 1

Reviewer 1 Report

The article is focused on the correlation between expression of TRIM21 in HNSCC cancers and survival chances to future propose TRIM21 as a prognostic marker.

The cohort selected by authors was composed of 306 male and 89 female patients. Can you explain why there is such imbalance between male and female?

It was not very clear that TRIM21 expression could be a consequence of immunological reaction as tumor develops. In this case, could be used as a prognostic marker in the case of the tumors with immune cell infiltration? As authors indicated, in the case of other malignancies,TRIM 21 behaviour is variable and it is very hard to be considered as a very reliable prognostic marker.

It is any correlation between TRIM21 expression and the treatment (chemotherapy/radiotherapy)?

Which is the level of expression of TRIM21 in normal tissue? Are any significant differences between normal and tumoral tissue regarding the expression of TRIM21 (inflammatory status etc)?

Reviewer 2 Report

This is an interesting and well-written manuscript, only minor changes are needed.

- Introduction: Please update ref. N#1 citing 1 or 2 of the following articles: 10.3390/cancers14246079, 10.1177/00034894211037194, 10.6004/jnccn.2020.0031, 10.1016/j.phrs.2021.105866.

- Discussion: Please cite and discuss the following articles: 10.3892/etm.2022.11697, 10.1186/s12929-020-0625-7.

- Materials and methods:

   - What about the histotypes? Are they all conventional SCC?

   - The authors state that "the characteristics and clinical data were obtained through the review of clinical records and pathology reports". Do they mean histological variables as well, such as depth of invasion, tumor budding, LVI, PNI? Weren't they included in the pathology reports, or could not they be retrieved upon revision of the slides? Since these features may be of prognostic significance, why were they not taken into account in the statistical analysis?

- IHC: Please mention positive and negative controls.

- The authors used the p16 status as a surrogate marker to assess the HPV status of their tumor samples. However, according to the latest WHO Classification of Head and Neck Tumors (5th Ed), P16 overexpression should not be considered a surrogate for HPV infection in hypopharingeal and laryngeal SCC, hence the authors should add a reference to support their statement.

Round 2

Reviewer 1 Report

Thank you for making the suggested revisions. I have no further comments.